# Towards a Sustainable World: Diversity of Freshwater Gastropods in Relation to Environmental Factors—A Case in the Konya Closed Basin, Türkiye

**Burçin Aşkım Gümüş** [1,*] **, Pınar Gürbüzer** [2] **and Ahmet Altındağ** [3]

1    Department of Biology, Faculty of Science, Gazi University, 06500 Ankara, Türkiye
2    Department of Hydrobiology, Faculty of Fisheries, Sinop University, 57000 Sinop, Türkiye
3    Department of Biology, Faculty of Science, Ankara University, 06100 Ankara, Türkiye
*    Correspondence: burcinaskim@gazi.edu.tr

**Abstract:** The Konya Closed Basin (KCB) in Türkiye plays a key role in agricultural production and freshwater supply. However, the basin is impacted by humanly derived nitrogenous compounds and toxic metals. Keeping the water quality at a potable level in the basin is compulsory. This study was part of a project yielding monitoring of water quality in KCB in accordance with the Water Framework Directive (2000/60/EC). Eleven stations, except Beyşehir Lake and Mamasın Dam, were sampled for the first time for freshwater molluscs. Community structure indexes and multivariate statistical analyses were applied to determine the microhabitats of gastropods and their responses to environmental changes. The structure and distribution of gastropod assemblages differed depending on total phosphate, total nitrogen, dissolved oxygen, and pH. This study revealed that most of the gastropods in KCB are relatively tolerant to biodegradable pollution. However, there is a strong observed decline in population size requiring intensive future monitoring; measures have to be taken to preserve the remaining populations. Two endemic species need an urgent action plan to protect their habitats: *Theodoxus anatolicus* of Çeltik Canal and *Bithynia pseudemmericia* of Beyşehir Lake; a re-assessment of their extinction risk according to the IUCN rules is needed (2022). The results of this study will be useful for comparison with future studies to document potential improvements or continued ecological regression in the quality of aquatic ecosystems in the watershed.

**Keywords:** freshwater; gastropod diversity; environmental factors; Konya Closed Basin; Türkiye

## 1. Introduction

The Anatolian freshwater malacofauna consists of 40 bivalve and 164 gastropod species with a high level of endemism (63%) [1]. It should be noted that much of the knowledge on the freshwater molluscan fauna is unfortunately out of date, with only a few recent reviews for Türkiye available [1–3]. There are major information gaps in species distribution, abundance, and size of the populations, and the threats to them. Seddon et al. [4] reported that there are a great number of species which are highly threatened or even extinct in the region. They determined water pollution, physical loss of wetlands, and natural system modifications due to human activities as the major threats to the freshwater malacofauna of Türkiye. Subsequently, field surveys and taxonomic studies are essential for in situ monitoring and conservation of the species. Unfortunately, we do not have a database on molluscs, such as MolluscaBase, in Türkiye for comparative studies [5]. Thus, we urgently need a database about freshwater molluscs in our country to initiate monitoring, conservation, and management plans at a legislative level. The preparation of an IUCN Red List facilitating the recognition of endangered species is also a prerequisite for protecting molluscs and their habitats [6]. The good news is that there is an increase in research of malacofauna diversity, seasonal changes in population structures of the freshwater molluscs and their interactions with environmental variables in situ: the

bioaccumulation of toxic chemicals ex situ and water quality determinations based on molluscs in Türkiye [7–11].

Our study is part of the Project "Monitoring of Water Quality in Konya Closed Basin (KCB) in the scope of Water Framework Directive (WFD)" led and funded by the Ministry of Forestry and Water Affairs, Directorate of Water Management, Department of Monitoring and System of Water Database, Republic of Türkiye [12]. The purpose of monitoring the KCB was to take measurements at the selected monitoring points and analyse some biological and physical parameters in river and lake waters that were deemed important by the ministry in accordance with the national legislation and Annex-5 of Article 8 of the WFD (2000/60/EC). The project plays an important role because it is the first multi-disciplined in situ study in which water quality assessments were applied in such a large basin in Türkiye. We collected, identified, counted, and catalogued molluscs as part of the project in 2014. The results were finally submitted to the ministry [12]. At that time, it was found that bivalve specimens were low in numbers (*Dreissena polymorpha*, 23 individuals; *Pisidium amnicum*, 10 individuals; and *Unio* sp., 2 individuals), and the distribution of bivalves was unbalanced between the localities. For this reason, it was decided that during the second phase of the study, the focus was shifted to gastropods, and a detailed assessment of population sizes with a subsequent comparison of literature data was performed. Currently, there is no study about the KCB water quality using indices based on gastropods.

Freshwater systems have important multi-usage components depending on community structure, the presence of invasive species, and water quality. The quality of the water is evaluated using indexes. Regarding the overall ecosystem health, the water quality should reach potable or at least palatable levels [13,14]. Currently, freshwater habitats worldwide experience massive negative impacts triggered by human activities, which have been summarized under the keywords, climate change or global warming, since the 1850s. The main consequences of these geophysical changes are massive and include the melting of polar ice, changes in the temperature regime, rising sea levels, increasing aridisation through increased evaporation, warming of surface waters, increasing hurricane and tornado activity with heavy rainfall and flooding, and much more. All these effects ultimately lead to a decrease in the storage of the continental waters, irreversible alterations in the ecological conditions, and finally major biodiversity losses [15,16].

Molluscs are vital inhabitants of the freshwaters. The ecosystem service of molluscs to the microbenthic fauna is multifold: the group is essential for the overall carbonate cycle and thus, molluscs are important ecosystem engineers. Gastropoda are detritus feeders, so they structure the aquatic bottom environments, while the filter feeding Bivalvia are essential for cleaning large amounts of water. Mollusc shells serve as substrate for hard-bottom dwellers in muddy environments and protect smaller invertebrates against predation. Another major role, however, is the production of enormous amounts of protein that serve as food for a large number of other freshwater taxa. Due to their limited dispersal patterns, their usually large populations and body sizes, and thanks to the relatively simple collection technique, easy identification, and ex situ processes, molluscs are widely used in biomonitoring programs [17]. They are highly sensitive to transformations of the environment and indicate changes to the ecological structure of a given habitat, to biological productivity, to water quality and biodegradation. A major feature is their susceptibility to heavy metal contamination. One of the largest molluscan classes, the gastropods, contains numerous species in aquatic environments. Often, these inhabit special microhabitats, and summarise environmental changes due to their long maturation times, low fecundity, and comparative longevity. They immediately react to human-mediated threats, including habitat loss, invasive species, and global warming. They are well known for having high levels of heavy metals in their bodies [18]. Thus, gastropods serve in environmental risk assessments and monitoring as pollution indicators for different compounds [19]. The initial response to heavy metal pollution in gastropods is declining abundance due to sexual abnormalities, termed the imposex, and super female phenomenon [20,21]. Toxic

chemical bioavailability in the shell, respiratory organ, visceral mass, and gonads qualifies gastropods for in situ and ex situ research [22–24].

The Konya Closed Basin is an important area in terms of its agricultural production. It ranks first in the production of barley, wheat, sugar beet, sunflower, tulip, cherry, dried beans, and carrots in Türkiye [25]. The region suffers from severely reduced levels of groundwater because of excessive water withdrawal for domestic and industrial use and prolonged drought periods due to climate change. These factors seriously threaten the sustainability of the agricultural potential and represent a major risk for the local biodiversity. The water quality in the basin was reported as class III–IV according to WFD (2000/60/EC) and the Turkish Surface Water Quality Regulation [26]. The major pollution sources in KCB are nitrogenous compounds and heavy metals derived from agricultural, industrial, and stockbreeding activities, as well as from urban wastewaters [27]. KCB is deeply structured by the presence of different types of surface freshwater ecosystems, such as wells, streamlets, rivers, or lakes. For example, it harbours Beyşehir Lake, which is the largest lake and freshwater reservoir in Türkiye, and one of the most important freshwater habitats in the Eastern Mediterranean Basin. Although declared a national park in 1993, it has still suffered from intense water abstraction in the last decades [28,29].

In the study presented here, our basic purpose was to investigate the species richness, diversity, and population size of the molluscs living at each sampling station, including seasonal effects. We used several biodiversity indices to describe the conditions under which the species were living. Amongst them, we determined their preferred water system, the overall trophic conditions, and their response to organic and chemical contamination. At the early stages of the project, the scientific board planned to study the phytoplankton, phytobenthos, macrophytes, macroinvertebrates, and fish species, excluding molluscs. However, our preliminary fieldwork revealed that KCB harbours endemic and locally restricted species of freshwater molluscs, which subsequently led us to include malacofaunistic studies into the project. The heavy metal contamination in the aquatic ecosystems of KCB is well documented. Over the last two decades, heavy metals, such as Cd, Co, Cr, Cu, Fe, Hg, Mg, Mn, Ni, Pb, Sr, Zn, and organic compounds, such as N, P, and NH4, have been studied in the waters, sediments, planktons, crabs, and fishes of the Mamasın Dam and Beyşehir Lake. Extreme levels of nutrient and toxic trace elements have been reported in these ecosystems, which are still important sources of potable water and fisheries in KCB [30–33].

According to Seddon et al. [4], the major threats beyond Threatened and Near Threatened freshwater molluscs are abstraction for domestic supplies, agriculture, and dams (68.8%), water pollution from agricultural and urban areas (56.3%), climate change leading to increasing droughts (29.7%), and loss of habitats due to urban (20.3%) and agricultural expansion (14.1%). The lakes, dams, and streams in KCB are currently shallow and eutrophic ecosystems [12]. They have been under pressure from heavy metal and nutrient loads, excessive water withdrawal, and habitat deterioration for decades [31]. In the long term, climate change will add to the anthropogenic alterations in the hydrological regime and turn KCB into a steppe or subdesert. The results we present should be used for further studies towards conservation not only of freshwater gastropods, but also the other elements of the biological community in the aquatic ecosystems of KCB.

## 2. Materials and Methods

### 2.1. Study Area

KCB is located in the Central Anatolian Region between 36°51′ and 39°29′ N, and 31°36′ and 34°52′ E. Its surface corresponds to 7% of the area of Türkiye. The coordinates of each sampling station were recorded using GPS (Garmin, etrex10). Table 1 contains the type, name, coordinates, and altitude of the sampling stations. The localities can be seen in Figure 1. The stations in the study area were marked using the Google Earth Pro program (V 9.171.0.0) and the satellite image was produced.

**Table 1.** Sampling stations in KCB.

| Sampling Station | SN * | Type | Coordinates | Altitude (m) |
|---|---|---|---|---|
| Çeltik Canal | 1 | Lotic | 36.352149° S; 420.8687° W | 1113 |
| Sarısu Stream | 2 | Lotic | 36.388649° S; 417.4620° W | 1139 |
| Üstünler Stream | 3 | Lotic | 36.375542° S; 416.4616° W | 1123 |
| Peçeneközü Stream | 4 | Lotic | 36.547932° S; 431.1860° W | 969 |
| Konya City Center Stream | 5 | Lotic | 36.447585° S; 419.0054° W | 1136 |
| Aksaray Input | 6 | Lotic | 36.579699° S; 424.5333° W | 951 |
| Beyşehir Lake | 7 | Lentic | 36.380361° S; 417.7373° W | 1127 |
| Suğla Reservoir | 8 | Lentic | 36.409740° S 413.1782° W | 1116 |
| Mamasın Dam | 9 | Lentic | 36.599326° S; 425.1335° W | 1051 |
| İvriz Dam | 10 | Lentic | 36.602412° S; 414.4768° W | 1127 |
| İbrala Dam | 11 | Lentic | 36.536539° S; 411.6160° W | 1070 |
| Akkaya Dam | 12 | Lentic | 36.641739° S; 419.8598° W | 1191 |
| Gödet Dam | 13 | Lentic | 36.527060° S; 410.6125° W | 1110 |

* SN: Station number; the station names were given without the type of the system in the following text. Modified after Altındağ et al. [12].

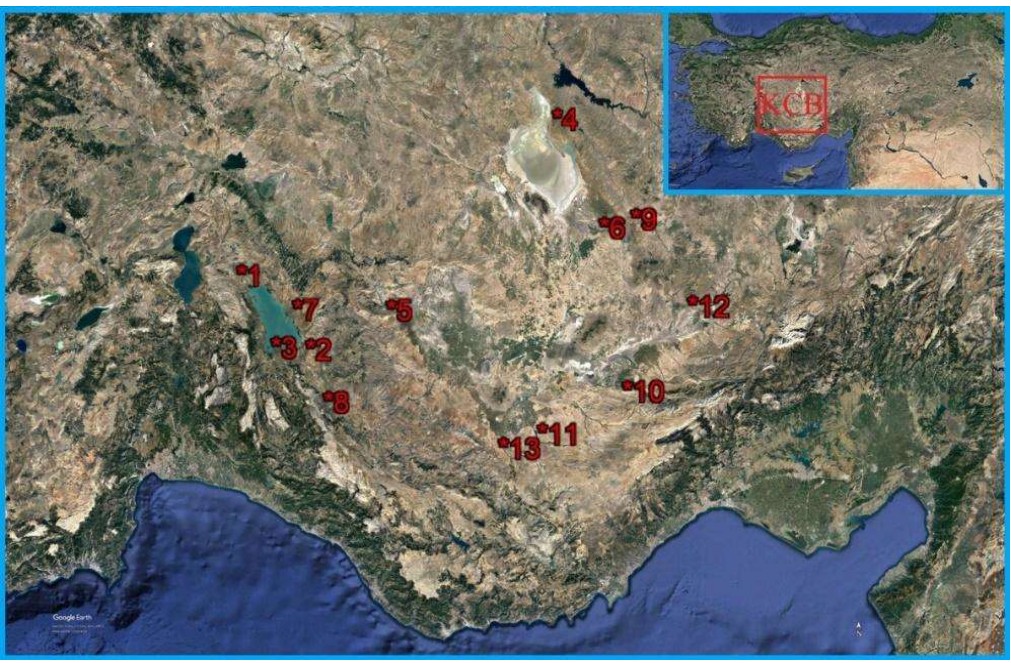

**Figure 1.** Sampling sites in KCB [station numbers (SN *) given in Table 1; produced by Burçin Aşkım Gümüş by using Google Earth Pro, V 9.171.0.0, 2022].

### 2.2. Collection and Identification of Gastropods

Within the scope of the project, the ecological status of 13 water bodies in KCB in accordance with the chemical, hydro-morphological, and biological quality elements according to the WFD was determined. The purpose of the project was to determine the quality of water in the selected sampling sites by using ecological multi-proxies. From this point of view, we primarily did not aim to detect the presence and abundance of the mollusc species at risk, or to monitor the mollusc survival and growth after relocation as described by Mackie et al. [34], but to detect the molluscan diversity in accordance with the ecological parameters of each sampling site at each sampling season. In this study, we surveyed 11 of the 13 KCB localities for the first time for molluscs. Molluscs were collected

at each site in the spring, summer, and fall seasons. We followed the recommendations of TS EN 28265: 2012 (TSQWR), ISO/TC 147/SC 6/WG 1, 3: 1982, and ISO 10870: 2012 in the sampling and preservation of molluscs [26,35,36]. We applied a combined method of qualitative and quantitative sampling according to Cummings et al. [37]. The methods used to survey each site differed depending on the water flow, water depth, and type of the substrate: (1) visual and tactile sampling of sediments within quadrats ($1 \times 1$ m) using hand zooplankton nets and grabs excavating the soft or hard bottom substrate; (2) sieving the substrate or sediment with griddles (mesh size: 10 mm, 5 mm, 2.5 mm, and 0.5 mm); and (3) handpicking the molluscs with pincers. At lotic sites, where the water current was medium fast and the bottom was stony, a metal framed and rectangular net with a 0.5 mm mesh size and a Ponar grab were used within quadrats to excavate the substrate, then the substrate was sieved with griddle sets. At lentic sites, where the water was deeper than >50 cm, stagnant, and muddy, an Ekman-Birge grab was used to dredge the soft substrate, and then the substrate was sieved with griddle sets. We collected molluscs from the supralittoral and littoral zones of shallow lotic and slowly moving lentic sites using the combined method described above (Figure 2). Live specimens were preserved in glass bottles with 70% ethyl alcohol, and dry shells were preserved in plastic boxes; all specimens were labelled with the sampling site data. We identified the specimens in the laboratory according to Schütt [38], Welter-Schultes [39], and Glöer [40] at species level. The numbers of identified specimens (NISP) were recorded.

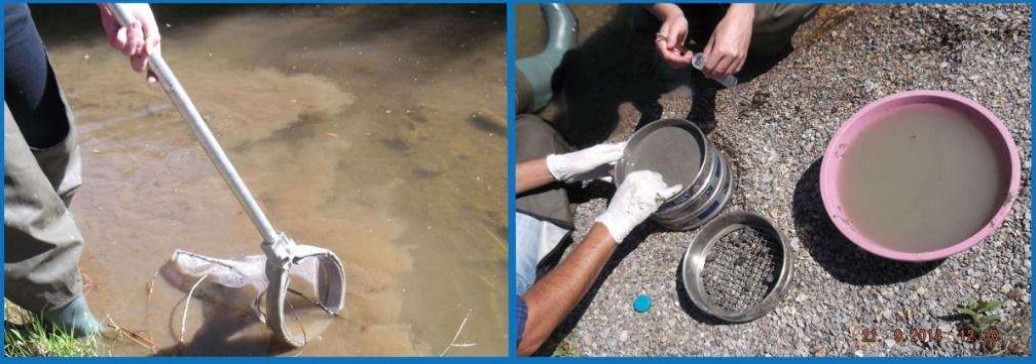

**Figure 2.** On the **left**: collecting gastropods with hand zooplankton net in Üstünler Stream; on the **right**: collecting gastropods with girdles, pincers, and by hand in Suğla Reservoir; photographed by Ahmet Altındağ on September 2014.

### 2.3. Counting Gastropods

We collected the specimens from the sampling sites within quadrats of $1 \times 1$ m. We used 39 quantitative samples (thirteen sites × three seasons), for a total of 531 identified specimens. We counted the number of each identified specimen at the species level.

### 2.4. Ecological Evaluation

The ecological interpretations were made based on the ecological preferences of the dominant gastropod species. According to Mariotte [41], the dominant species of each sampling station was determined with a cumulative relative abundance of >12%. The relative abundance was calculated as the mean value of the ratio of the individual number of the species and the total individual number of all recorded species in each site. We applied the formalised data of dominant species' niches and sensitivity levels to pollution used by Mouthon and Charvet [42], and Hubenov [43]. Afterwards, we compared the existence or absence of gastropods with the current water quality of the sampling sites determined in the project [12].

### 2.5. Environmental Variables

We applied the ISO/TC 147/SC 5, EN ISO 8689-1, and EN ISO 8689-2 standards in selecting the ecological parameters [44–46]. In situ measurements were executed at every sampling site during each sampling season. Temperature, electrical conductivity, dissolved oxygen, and pH were measured by a multi-parameter water quality sensor, and Secchi disc depth was measured. Water samples were collected into 1 LT glass bottles from each site at each sampling season to analyse the total nitrogen, total phosphorus, total kjeldahl nitrogen, nitrite, and nitrate values in the laboratory. The methods and devices used in measuring the environmental parameters are given in Table 2.

**Table 2.** Methods used in measuring the environmental parameters (LOD: limit of detection); modified after Altındağ et al. [12].

| Parameter | Method | Device | LOD |
| --- | --- | --- | --- |
| Temperature (°C) | SM 2550 B | Multi-parameter water quality sensor (MPS) | - |
| pH | TS EN ISO 10523 | MPS | - |
| Electrical conductivity ($\mu$S c$^{-1}$) | TS 9748 EN 27888 | MPS | - |
| Dissolved oxygen (mg L$^{-1}$ O$_2$) | TS EN 5814 | MPS | - |
| Light transmittance (m) | EPA Volunteer Stream | Secchi disc | - |
| Chlorophyll a (mg L$^{-1}$) | SM 10200 H | Spectrophotometer | 0.001 |
| Total nitrogen (mg L$^{-1}$ N) | SM 4500 NO$_2$ B<br>SM 4500 Norg B<br>EPA METHOD 352-1 | -Nitrogen-Protein device<br>-Spectrophotometer<br>-Hot plate | 0.2 |
| Total phosphorus (mg L$^{-1}$ P) | SM 4500 P B E | -Spectrophotometer<br>-Hot plate | 0.01 |
| Total kjeldahl nitrogen (mg L$^{-1}$) | SM 4500 Norg B | Nitrogen-Protein device | 0.2 |
| Nitrite (mg NO$_2{}^-$-N L$^{-1}$) | SM 4500 NO2 B | Spectrophotometer | 0.002 |
| Nitrate (mg NO$_3{}^-$-N L$^{-1}$) | EPA METHOD 352-1 | -Spectrophotometer<br>-Hot plate | 0.1 |

### 2.6. Statistical Methods

Before the analysis, all data (except pH) were log10 (x + 1) transformed to approximate normal distribution. We used the CANOCO 4.5 package program [47] to determine relationships between ecological parameters and species abundances. According to detrended correspondence analysis (DCA), the longest gradient of the lotic ecosystem was the second, and the lentic ecosystem was the first axis. The lengths of the gradients were 7.313 and 9.981, respectively. They were larger than 4.0, and the data showed a unimodal distribution. Therefore, we chose canonical correspondence analysis (CCA) for multivariate analysis [48]. A PRIMER5 package program [49] was used to apply non-metric multidimensional scaling (nMDS), cluster, analysis of similarity (ANOSIM), and similarity percentages (SIMPER) based on the Bray-Curtis similarity coefficient [50]. PRIMER5 was used for analysis using the Shannon-Weaver diversity index [51], the Margalef species richness index [52] and Pielou's evenness index [53]. The frequency analyses of taxa were evaluated by Soyer's [54] frequency index. Bellan-Santini's [55] method was applied for calculating the abundance of taxa.

## 3. Results

### 3.1. In General

A total of 21 freshwater gastropod species (two Neritimorpha, four Caenogastropoda, and 15 Heterobranchia) were identified (NISP: 531). The species list and the distribution of gastropods in sampling sites were given in Table 3. The number of identified specimens

(individual m$^{-2}$) of lotic and lentic sites as the sum of three sampling seasons were given in Tables 4 and 5. *Theodoxus fluviatilis*, *Melanoides tuberculata*, *Valvata saulcyi*, *Gyraulus albus*, and *Ampullaceana balthica* from Beyşehir, and *Potamopyrgus antipodarum* from Sarısu and Peçeneközü were recorded for the first time in KCB. The lotic sites were inhabited by Caenogastropoda (91 individuals m$^{-2}$), Pulmonata (81 individuals m$^{-2}$), Valvatidae (24 individuals m$^{-2}$), and Neritidae (7 individuals m$^{-2}$) (Table 4). The lentic sites were inhabited by Pulmonata (195 individuals m$^{-2}$), Caenogastropoda (105 individuals m$^{-2}$), Valvatidae (15 individuals m$^{-2}$), and Neritidae (13 individuals m$^{-2}$) (Table 5). In brief, we determined that the prosobranch gastropods dominated the lotic sites, while pulmonates dominated the lentic sites. The present research was not designed in the field of taxonomy, but we preferred using the names *Peregriana* and *Ampullaceana* following the molecular revision of Radicine gastropods by Aksenova et al. [56]. However, different points of view do exist beyond taxonomists. The species and subclass names in Table 3 are the authoritative molluscan names approved by the scientific community in the editorial board of MolluscaBase [5].

**Table 3.** The species list and the distribution of gastropods in sampling sites.

| Species | Lotic Sites | Lentic Sites |
| --- | --- | --- |
| *Theodoxus anatolicus* (Récluz, 1841) [NER (PRO)] | C | $\phi$ |
| *Theodoxus fluviatilis* (Linnæus, 1758) [NER (PRO)] | C, S | B |
| *Viviparus viviparus* (Linnæus, 1758) [CAE (PRO)] | AI | SR, M |
| *Melanoides tuberculata* (O.F. Müller, 1774) [CAE (PRO)] | $\phi$ | B |
| *Bithynia pseudemmericia* Schütt, 1964 [CAE (PRO)] | $\phi$ | B |
| *Potamopyrgus antipodarum* (J.E. Gray, 1843) [CAE (PRO)] | S, P | $\phi$ |
| *Valvata piscinalis* (O.F. Müller, 1774) [HET (PRO)] | AI | B |
| *Valvata saulcyi* Bourguignat, 1853 [HET (PRO)] | $\phi$ | B |
| *Galba truncatula* (O.F. Müller, 1774) [HET (PUL)] | K | A |
| *Stagnicola palustris* (O.F. Müller, 1774) [HET (PUL)] | AI | A |
| *Radix auricularia* (Linnæus, 1758) [HET (PUL)] | $\phi$ | SR |
| *Peregriana labiata* (Rossmässler, 1835) [HET (PUL)] | U, AI | B, SR, M, I |
| *Ampullaceana balthica* (Linnæus, 1758) [HET (PUL)] | C, S, K | B, SR, M, I, IB |
| *Lymnaea stagnalis* (Linnæus, 1758) [HET (PUL)] | $\phi$ | B, G |
| *Physa fontinalis* (Linnæus, 1758) [HET (PUL)] | $\phi$ | SR, M |
| *Physella acuta* (Draparnaud, 1805) [HET (PUL)] | C, U, P, AI | M |
| *Planorbarius corneus* (Linnæus, 1758) [HET (PUL)] | $\phi$ | B, SR |
| *Planorbis planorbis* (Linnæus, 1758) [HET (PUL)] | $\phi$ | B, A |
| *Bathyomphalus contortus* (Linnæus, 1758) [HET (PUL)] | C | $\phi$ |
| *Gyraulus albus* (O.F. Müller, 1774) [HET (PUL)] | S, K | B, IB, G |
| *Gyraulus piscinarum* (Bourguignat, 1852) [HET (PUL)] | S, K | B, IB |

NER: Neritimorpha; CAE: Caenogastropoda; HET: Heterobranchia; (PRO): formerly Prosobranchia; (PUL): formerly Pulmonata; $\phi$: not existing; lotic sites- AI: Aksaray; C: Çeltik; U: Üstünler; P: Peçeneközü; S: Sarısu; K: Konya Center; lentic sites-A: Akkaya; IB: İbrala; I: İvriz; M: Mamasın; SR: Suğla; G: Gödet; B: Beyşehir.

When Cluster and nMDS were applied to all the data, no clear separation was observed in all sites. However, two similar groups were identified in the lotic sites (Figure 3). The summer and fall seasons of Üstünler and Aksaray were clustered in one group, and Sarısu and Konya Centre were clustered in another group. Peçeneközü and Çeltik had low species diversity and differed within the seasons. Suğla and Mamasın had similar

—

species composition in the lentic sites in all seasons. It was also understood that Gödet and especially Akkaya had an excessively different species composition from other lentic sites and seasons according to the Cluster analysis. The results were supported by nMDS (Figure 4).

**Table 4.** Species list, number of identified specimen (individual m$^{-2}$) at each site.

| Species | Çeltik | Sarısu | Üstünler | Peçeneközü | Konya C. | Aksaray | Σind | D% | F% |
|---|---|---|---|---|---|---|---|---|---|
| Bcont | 1 | 0 | 0 | 0 | 0 | 0 | 1 | 0.49 | 16.67 |
| Gtrun | 0 | 0 | 0 | 0 | 1 | 0 | 1 | 0.49 | 16.67 |
| Galbu | 0 | 2 | 0 | 0 | 5 | 0 | 7 | 3.45 | 33.33 |
| Gpisc | 0 | 12 | 0 | 0 | 17 | 0 | 29 | 14.29 | 33.33 |
| Pacut | 2 | 0 | 16 | 1 | 0 | 6 | 25 | 12.32 | 66.67 |
| Panti | 0 | 12 | 0 | 53 | 0 | 0 | 65 | 32.02 | 33.33 |
| Rbalt | 1 | 3 | 0 | 0 | 30 | 0 | 34 | 16.75 | 50 |
| Rlabi | 0 | 0 | 1 | 0 | 0 | 4 | 5 | 2.46 | 33.33 |
| Spalu | 0 | 0 | 0 | 0 | 0 | 4 | 4 | 1.97 | 16.67 |
| Tfluv | 1 | 5 | 0 | 0 | 0 | 0 | 6 | 2.96 | 33.33 |
| Tanat | 1 | 0 | 0 | 0 | 0 | 0 | 1 | 0.49 | 16.67 |
| Vpisc | 0 | 0 | 0 | 0 | 0 | 24 | 24 | 11.82 | 16.67 |
| Vvivi | 0 | 0 | 0 | 0 | 0 | 1 | 1 | 0.49 | 16.67 |
| S | 5 | 5 | 2 | 2 | 4 | 5 | | | |
| N | 6 | 34 | 17 | 54 | 53 | 39 | Σ203 | | |
| d | 2.23 | 1.13 | 0.35 | 0.25 | 0.76 | 1.09 | | | |
| J′ | 0.97 | 0.87 | 0.32 | 0.13 | 0.71 | 0.71 | | | |
| H(loge) | 1.56 | 1.4 | 0.22 | 0.09 | 0.98 | 1.15 | | | |
| 1-λ′ | 0.93 | 0.74 | 0.12 | 0.04 | 0.58 | 0.59 | | | |

Σind: total specimen number of each species at all lotic sites; dominance (D%), frequency (F%), total species (S), total individuals (N), and Margalef Richness (d), Pielou's Evenness (J′), Shannon Diversity [H(loge)], and Simpson Diversity (1-λ′) indexes in lotic sites (Bcont: *B. contortus*, Gtrun: *G. truncatula*, Galbu: *G. albus*, Gpisc: *G. piscinarum*, Pacut: *P. acuta*, Panti: *P. antipodarum*, Rbalt: *A. balthica*, Rlabi: *P. peregra*, Spalu: *S. palustris*, Tfluv: *T. fluviatilis*, Tanat: *T. anatolicus*, Vpisc: *V. piscinalis*, Vvivi: *V. viviparus*).

### 3.2. In Lotic Stations

Thirteen species were identified (NISP: 203). The number of individuals and species was low (Table 4). When examining the diversity, low values with an even distribution were determined for all but Üstünler and Peçeneközü. Konya Center and Peçeneközü streams were rich in specimen numbers. *Potamopyrgus antipodarum* was the dominant (32.02%), while *Physella acuta* was the most frequent (66.67%) species.

### 3.2.1. Similarities of Lotic Stations

In addition to nMDS and Cluster analysis for further analysis of similarities, SIMPER was used, and Konya Centre showed no similarity with Aksaray, Üstünler and Peçeneközü. Sarısu had no similarity with Üstünler and Aksaray. Evaluating similarities, we determined notably low values in between (maximum similarity was 45.41% between Peçeneközü and Üstünler). The results of dissimilarities amongst lotic sites were supported by ANOSIM (Global test, R = 0.403, *p* < 0.05) and nMDS biplot (Figure 4). The Aksaray has a very different species composition among the other lotic sites in the basin, which is shown in the CCA triplot (Figure 5).

**Table 5.** Species list, number of identified specimen (individual m$^{-2}$).

| Species | Beyşehir | Suğla | Mamasın | İvriz | İbrala | Akkaya | Gödet | Σind | D% | F% |
|---|---|---|---|---|---|---|---|---|---|---|
| Bpseu | 7 | 0 | 0 | 0 | 0 | 0 | 0 | 7 | 2.13 | 14.29 |
| Gtrun | 0 | 0 | 0 | 0 | 0 | 5 | 0 | 5 | 1.52 | 14.29 |
| Galbu | 7 | 0 | 0 | 0 | 2 | 0 | 4 | 13 | 3.96 | 42.86 |
| Gpisc | 3 | 0 | 0 | 0 | 2 | 0 | 0 | 5 | 1.52 | 28.57 |
| Lstag | 2 | 0 | 0 | 0 | 0 | 0 | 1 | 3 | 0.91 | 28.57 |
| Mtuber | 4 | 0 | 0 | 0 | 0 | 0 | 0 | 4 | 1.22 | 14.29 |
| Pfont | 0 | 16 | 8 | 0 | 0 | 0 | 0 | 24 | 7.32 | 28.57 |
| Pacut | 0 | 0 | 2 | 0 | 0 | 0 | 0 | 2 | 0.61 | 14.29 |
| Pcorn | 1 | 8 | 0 | 0 | 0 | 0 | 0 | 9 | 2.74 | 28.57 |
| Pplan | 1 | 0 | 0 | 0 | 0 | 4 | 0 | 5 | 1.52 | 28.57 |
| Rauri | 0 | 5 | 0 | 0 | 0 | 0 | 0 | 5 | 1.52 | 14.29 |
| Rbalt | 77 | 4 | 3 | 3 | 5 | 0 | 0 | 92 | 28.05 | 71.43 |
| Rlabi | 14 | 2 | 2 | 1 | 0 | 0 | 0 | 19 | 5.79 | 57.14 |
| Spalu | 0 | 0 | 0 | 0 | 0 | 13 | 0 | 13 | 3.96 | 14,29 |
| Tfluv | 13 | 0 | 0 | 0 | 0 | 0 | 0 | 13 | 3.96 | 14.29 |
| Vpisc | 12 | 1 | 0 | 0 | 0 | 0 | 0 | 13 | 3.96 | 28.57 |
| Vsaul | 2 | 0 | 0 | 0 | 0 | 0 | 0 | 2 | 0.61 | 14.29 |
| Vvivi | 0 | 43 | 51 | 0 | 0 | 0 | 0 | 94 | 28.66 | 28.57 |
| S | 12 | 7 | 5 | 2 | 3 | 3 | 2 | | | |
| N | 143 | 79 | 66 | 4 | 9 | 22 | 5 | Σ328 | | |
| d | 2.22 | 1.37 | 0.95 | 0.72 | 0.91 | 0.65 | 0.62 | | | |
| J′ | 0.66 | 0.7 | 0.5 | 0.81 | 0.91 | 0.87 | 0.72 | | | |
| H(loge) | 1.65 | 1.36 | 0.81 | 0.56 | 1 | 0.96 | 0.5 | | | |
| 1-λ′ | 0.68 | 0.65 | 0.39 | 0.5 | 0.67 | 0.59 | 0.4 | | | |

Σind: total specimen number of each species at all lotic sites; specimen number of each species at all lentic sites; dominance (D%), frequency (F%), total species (S), total individuals (N), and Margalef Richness (d), Pielou's Evenness (J′), Shannon Diversity [H(loge)], and Simpson Diversity (1-λ′) indexes in lentic sites (Bpseu: *B. pseudemmericia*, Gtrun: *G. truncatula*, Galbu: *G. albus*, Gpisc: *G. piscinarum*, Lstag: *L. stagnalis*, Mtuber: *M. tuberculata*, Pfont: *P. fontinalis*, Pacut: *P. acuta*, Pcorn: *P. corneus*, Pplan: *P. planorbis*, Rauri: *R. auricularia*, Rbalt: *A. balthica*, Rlabi: *P. peregra*, Spalu: *S. palustris*, Tfluv: *T. fluviatilis*, Vpisc: *V. piscinalis*, Vsaul: *V. saulcyi*, Vvivi: *V. viviparus*).

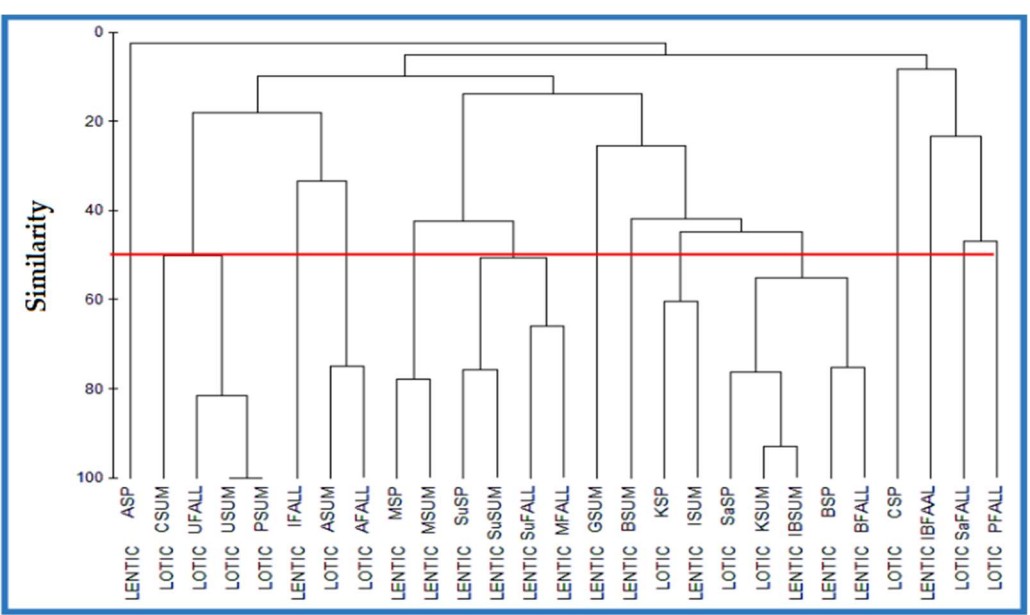

**Figure 3.** Cluster analysis of similarities in both lotic and lentic sites (lotic sites-A: Aksaray; C: Çeltik; U: Üstünler; P: Peçeneközü; Sa: Sarısu; K: Konya Center; lentic sites-A: Akkaya; IB: İbrala; I: İvriz; M: Mamasın; Su: Suğla; G: Gödet; B: Beyşehir; SP: Spring; SUM: Summer; FALL: Autumn; red line: below 50% similarity).

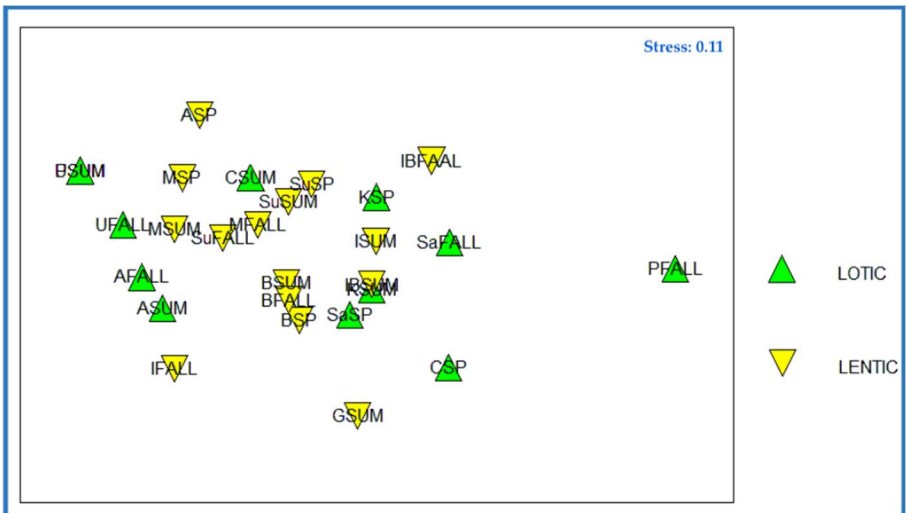

**Figure 4.** nMDS results of lotic and lentic sites (green triangles: lotic sites-A: Aksaray; C: Çeltik; U: Üstünler; P: Peçeneközü; Sa: Sarısu; K: Konya Centre; yellow triangles: lentic sites-A: Akkaya; IB: İbrala; I: İvriz; M: Mamasın; Su: Suğla; G: Gödet; B: Beyşehir; SP: Spring; SUM: Summer; FALL: Autumn).

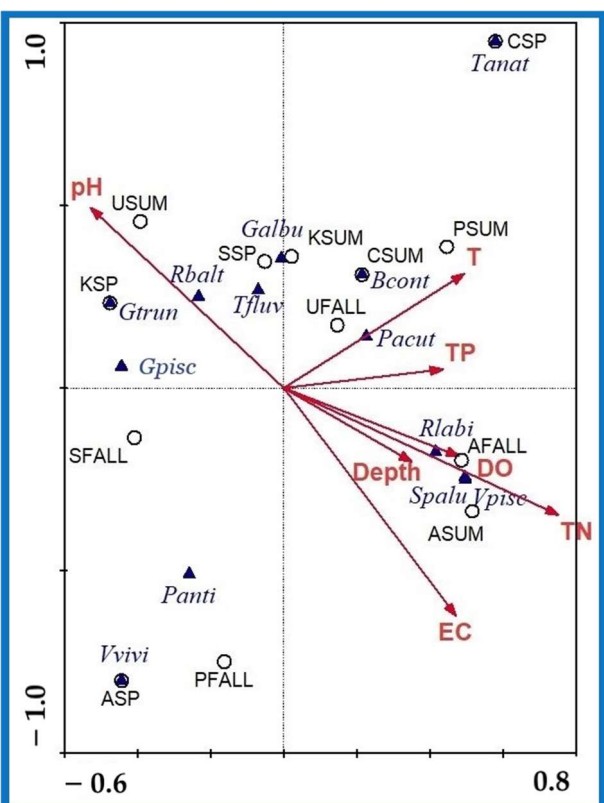

**Figure 5.** Canonical correspondence analysis triplot with lotic sites (red arrows: environmental parameters, black circles: sites, blue triangle: species; A: Aksaray, C: Çeltik, U: Üstünler, P: Peçeneközü, S: Sarısu, K: Konya Centre, SP: Spring, SUM: Summer, FALL: Autumn; Bcont: *B. contortus*, Gtrun: *G. truncatula*, Galbu: *G. albus*, Gpisc: *g. piscinarum*, Pacut: *P. acuta*, Panti: *P. antipodarum*, Rbalt, *A. balthica*, Rlabi: *P. peregra*, Spalu: *S. palustris*, Tfluv: *T. fluviatilis*, Tanat: *T. anatolicus*, Vpisc: *V. piscinalis*, Vvivi: *V. viviparus*; T: water temperature, Depth: water depth; TP: total phosphorus, TN: total nitrogen, DO: dissolved oxygen, EC: electrical conductivity).

### 3.2.2. CCA results of lotic stations

1. *Valvata piscinalis* and *A. balthica* had a positive correlation with total nitrogen.
2. *Theodoxus fluviatilis*, *G. albus*, *Gyraulus piscinarum*, *Galba truncatula*, *A. balthica*, with high frequency values had a positive correlation with alkalinity, and *P. acuta* and *Bathyomphalus contortus* with temperature.
3. *Viviparus viviparus* and *P. antipodarum* had a negative correlation with total phosphorus and total nitrogen.
4. No seasonal group was observed in the CCA analysis in lotic sites; it was also consistent with the nMDS and Cluster analysis results (Figures 3–5).

### 3.3. In Lentic Stations

Eight species were identified (NISP: 328). Species richness and diversity were variable. Beyşehir was the richest and most diverse station, with 143 specimens, and 12 species. *Viviparus viviparus* was the dominant (28.66%), *A. balthica* was the most frequent (71.43%) species (Table 5).

#### 3.3.1. Similarities of Lentic Stations

According to the SIMPER analysis, Suğla and Mamasın had similar species compositions (40.9%) in all seasons. The results were supported by the ANOSIM analysis (Global test, R = 0.636, $p < 0.01$) of lentic sites and the nMDS biplot (Figure 4).

#### 3.3.2. CCA Results of Lentic Stations

1. *Ampullaceana balthica* had a positive correlation with water temperature and Secchi disc transparency as shown at Figure 6.

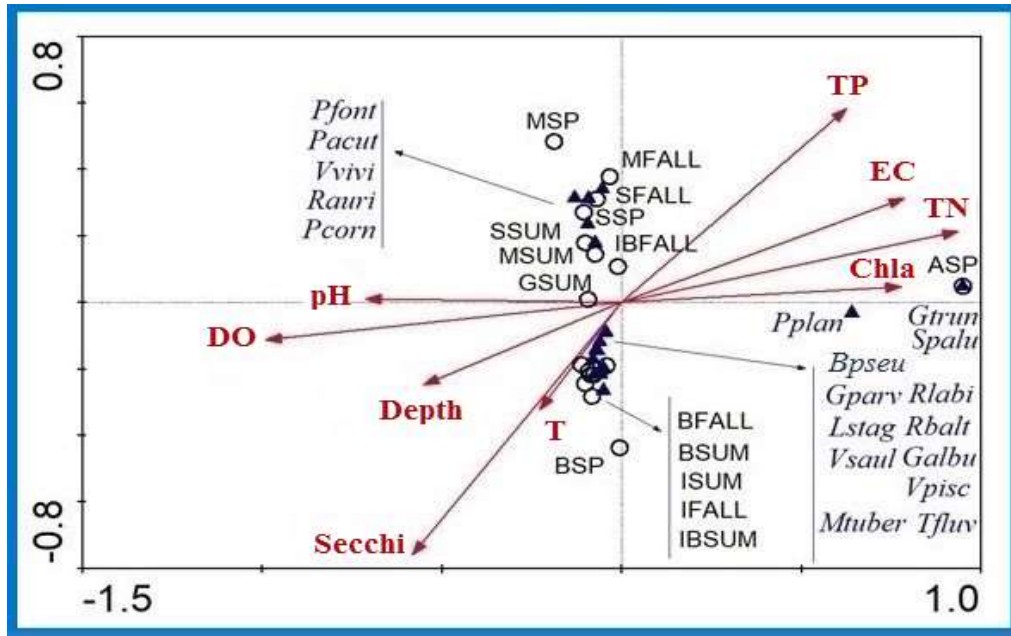

**Figure 6.** Canonical correspondence analysis triplot with lentic sites (red arrows: environmental parameters, black circles: sites, blue triangle: species; A: Akkaya, IB: İbrala I: İvriz, M: Mamasın, S: Suğla, G: Gödet, B: Beyşehir, SP: Spring, SUM: Summer, FALL: Autumn; Bpseu: *B. pseudemmericia*, Gtrun: *G. truncatula*, Galbu: *G. albus*, Gpisc: *G. piscinarum*, Lstag: *L. stagnalis*, Mtuber: *M. tuberculata*, Pfont: *P. fontinalis*, Pacut: *P. acuta*, Pcorn: *P. corneus*, Pplan: *P. planorbis*, Rauri: *R. auricularia*, Rbalt: *A. balthica*, Rlabi: *P. peregra*, Spalu: *S. palustris*, Tfluv: *T. fluviatilis*, Vpisc: *V. piscinalis*, Vsaul: *V. saulcyi*, Vvivi: *V. viviparus*; T: water temperature; Depth: water depth; Secchi: turbidity; TP: total phosphorus; TN: total nitrogen; DO: dissolved oxygen; EC: electrical conductivity; Chla: chlorophyll a).

2. *Planorbis planorbis*, *G. truncatula*, and *S. palustris* had a positive correlation with chlorophyll a, and total nitrogen.
3. Existence of *V. viviparus* and *Physa fontinalis* were positively correlated with increasing levels of pH and DO.
4. According to the CCA triplot, species and lentic sites clustered into three groups. The first group included all seasons of Beyşehir and İvriz, as well as the summer season of İbrala. The species composition of the first group had a positive correlation with temperature and Secchi disc. The second group consisted of only Akkaya and its species composition. The gastropod presence of Akkaya was directly correlated with chlorophyll a, and total nitrogen. The third group consisted of the spring and fall seasons of İbrala, and all seasons of Suğla, Mamasın, and Gödet. The species composition of the third group was located distantly from temperature and other nutrients (Figure 6).

## 4. Discussion

In our study, we reported 39 malacological surveys on 13 sites and identified 531 individuals at the species level. The low abundance of stenoecious gastropod species and the highly abundant (relative to other species in the community), uneven euryoecious species were significant for the eutrophic and hypertrophic conditions that dominated the waters of KCB. Even in the mesotrophic Beyşehir Lake with a 650 km$^2$ surface area, only 143 individuals m$^{-2}$ belonging to 12 species could be collected.

According to the results obtained from analyses of similarities and dissimilarities, we determined that the gastropod assemblages were variable in the waters of KCB. The species compositions differed even within sampling seasons in most of the sites (Figures 3–6). The lotic sites in KCB were shallow ecosystems (Çeltik: water depth was max. 50 cm; Sarısu: 44 cm depth; Üstünler: 50 cm depth; Peçeneközü: 22 cm depth; Konya Centre: 25 cm depth; Aksaray: 70 cm depth), and they were under pressure of long-term droughts, in addition water withdrawal [12]. We could not collect specimens from Çeltik and Konya Centre during autumn surveys, as they dried out completely. Sarısu and Üstünler are streams feeding Beyşehir Lake [12]. Despite the lake-stream networks, we determined that Sarısu, Üstünler, and Beyşehir differed in species compositions. Therefore, we decided to discuss the results of this study from the point of the dominant species' niches and tolerance to environmental variables at each sampling site in KCB.

The names of dominant species, quantity, relative abundance (D%), niche, and sensitivity level to pollution in accordance with Mouthon and Charvet [42], and Hubenov [43], the diversity (WFD), trophic (TSWQR) and chemical structures (EQS) of each site are given in Table 6 [12]. Unfortunately, 4 of 13 sites were classified as hypertrophic (Üstünler, Aksaray, İbrala, Akkaya), 8 sites were eutrophic (Çeltik, Sarısu, Peçeneközü, Konya Center, Suğla, Mamasın, İvriz, Gödet), and 1 site was mesotrophic (Beyşehir) [12]. The habitat preferences of the dominant species were compatible with the current trophic structure of each site.

Freshwater gastropods, whether they have ctenidia or rudimentary respiratory organs, prefer alkaline waters with high levels of C, Ca, and Mg. *Gyraulus albus*, *G. piscinarum*, and *A. balthica* can live in a wide range of pH. *Ampullaceana balthica*, and *G. truncatula* can live in thermal waters [42,43]. The presence of *A. balthica*, and *G. piscinarum* in Konya Center, *G. piscinarum* in Sarısu, and *P. acuta* in Üstünler was found compatible with their tolerance range for pH and temperature (Tables 4 and 6, Figure 5). Beyşehir, İvriz and İbrala were inhabited by the highly adaptive *A. balthica*. We observed that the mesotrophic conditions of Beyşehir increased the population size of *A. balthica* (77 inds. m$^{-2}$) (Tables 5 and 6, Figure 6) [42].

*Physella acuta* is an alien species with a high tolerance to heavy metal pollution and has been proposed as a pollution biomonitoring organism [57]. In this study, Çeltik and Üstünler were dominated by this species (2 inds. m$^{-2}$, and 16 inds. m$^{-2}$ respectively) (Table 4). Pb, Ni and trichloromethane were above the maximum allowable concentration (>mac) in Çeltik, and Pb, nonylphenols, and trichloromethane were above the maximum

allowable concentration (>mac) in Üstünler (Table 6) [12]. We observed that the total number of individuals was <20 in three seasons of Çeltik and Üstünler. The existence and dominance of *P. acuta* in heavy metal polluted and eutrophic Çeltik and hypertrophic Üstünler was consistent with the literature.

**Table 6.** Dominant species of each site, D%, tolerance level to pollution according to Mouthon and Charvet [42], and ecological data according to Hubenov [43].

| Station | WFD | TSWQR | EQS | Sp [D] | D% | MC [D] | Hub [D] |
|---------|-----|-------|-----|--------|-----|--------|---------|
| Çeltik | bad | Eutrophic | >mac | Pacut | 33 | 1 | eu, pe, tx, α-β |
| Sarısu | bad | Eutrophic | >mac | Gpisc | 35 | $\phi$ | sw, po, ph |
| | | | | Panti | 35 | 5 | eu, sw, po, is |
| Üstünler | poor | Hypertrophic | >mac | Pacu | 94 | | |
| Peçeneközü | poor | Eutrophic | >mac | Panti | 98 | | |
| Konya C. | bad | Eutrophic | >mac | Rbalt | 57 | $\phi$ | sw, pe, po, ph |
| Aksaray | poor | Hypertrophic | >mac | Vpisc | 62 | 7 | sw, pe, po, ph, rh, xs |
| Beyşehir | poor | Mesotrophic | >mac | Rbalt | 54 | | |
| Suğla | bad | Eutrophic | >mac | Vvivi | 54 | 8 | sw, pe, po |
| Mamasın | bad | Eutrophic | >mac | Vvivi | 77 | | |
| İvriz | bad | Eutrophic | >mac | Rbalt | 75 | | |
| İbrala | bad | Hypertrophic | >mac | Rbalt | 56 | | |
| Akkaya | bad | Hypertrophic | >mac | Spalu | 59 | 4 | eu, pe, ph |
| Gödet | bad | Eutrophic | >mac | Galbu | 80 | 5 | sw, ph, po, rh |

WFD: ASPT/BBI/BMWP indexes for macroinvertebrates according to EU-WFD; TSWQR: Trophic level according to Turkish Surface Water Quality Regulation; EQS: heavy metals (Pb, Ni, Al, As) and Nonylphenol, Trichloromethane, Benzo [a] pyrene) were >mac: above maximum allowable concentration according to Environmental Quality Standards of 2013/39/EU in Altındağ et al. [12]; Sp [D]: Dominant species; MC [D]: Pollution tolerance of dominant species (numbers between 1–5: tolerant; 6–9: increasing tolerance; 10–13: sensitive; $\phi$: no information) [42]; Hub [D]: Ecological data of dominant species; eu, eurybiont; is, invasive; pe, pelophilous; ph, phytophilous; po, potamophilous; rh, rhithrophilous; α-β, α-β-mesosaprobic; xs, xenosaprobic; sw, stagnant water; tx, trogloxene [43] (Pacut: *P. acuta*, Gpisc: *G. piscinarum*, Panti: *P. antipodarum*, Rbalt: *A. balthica*, Vpisc: *V. piscinalis*, Vvivi: *V. viviparus*, Spalu: *S. palustris*, Galbu: *G. albus*).

*Potamopyrgus antipodarum* known as the New Zealand mud snail is an invasive species and has been introduced to Türkiye [39]. It is an invasive gastropod favoured by the nitrate pollution of non-marine waters because it feeds on detritus and tolerates mild biodegradable and metal pollution [58]. In this study, we determined the dominance of *P. antipodarum* in lotic sites. It was found in Sarısu with 12 individuals m$^{-2}$ and Peçeneközü with 53 individuals m$^{-2}$ (Table 4). Sarısu and Peçeneközü were polluted by Pb, Ni and trichloromethane (Table 6), and the eutrophic conditions of these shallow, muddy clay benthic substrate streams supported the invasion of this species [12]. Although there were no specimens of *P. antipodarum* found in lentic sites, we think that the rapid spread of this gastropod will be harmful to the native molluscs of KCB in the future.

*Stagnicola palustris* lives in stagnant or slow running freshwaters on muddy substrate with flourishing macrophytes. Although it is highly tolerant to eutrophication, it was found low in numbers (4 inds. m$^{-2}$) in the hypertrophic conditions of Aksaray, and hardly reached 13 inds. m$^{-2}$ in the hypertrophic Akkaya (Tables 4–6) [42,43].

*V. viviparus* prefers clean locations without the pressure of pollutants such as nitrogen and phosphorus. It is a seston-feeder, lies for long periods in the mud with the mouth uppermost, and can even live in brackish waters. The survival of this prosobranch depends on high levels of dissolved oxygen [42,43]. In this study, one specimen of *V. viviparus* was

recorded representing the weakly oxygenated and hypertrophic waters of Aksaray (Table 4). The distribution of *V. viviparus* from the slightly eutrophic Suğla and Mamasın with high dominance values was compatible with their ecological demands (Tables 5 and 6) [42,43,59].

Freshwater prosobranch species are mostly stenoecious, while pulmonates are euryoecious [60]. The extreme conditions, such as oligotrophy and dystrophy, end up with a paucity of stenoecious and a dominance of euryoecious molluscs [61]. The low diversity of gastropods, and the predominance of tolerant species in KCB, were compatible with the eutrophic and hypertrophic conditions of the basin. The presence of *T. fluviatilis* (Çeltik-1 individual m$^{-2}$, Sarısu-5 inds. m$^{-2}$, Beyşehir-13 inds. m$^{-2}$), *V. viviparus* (Aksaray-1 ind. m$^{-2}$, Suğla-43 inds. m$^{-2}$, Mamasın-51 inds. m$^{-2}$), and even xenosaprobic *V. piscinalis* (Aksaray-24 inds. m$^{-2}$, Beyşehir-12 inds. m$^{-2}$, Suğla-1 inds. m$^{-2}$) in KCB seemed to be controversial with this ecological data. The literature, however, suggests that they are adaptable to changes in trophic circumstances [39,42,43,62], and the population numbers of these species were inversely linked with the degree of eutrophication at the sample sites (Tables 4–6, Figures 5 and 6).

As shown in Figure 7, even the most tolerant species could hardly survive in the hypertrophic conditions of Aksaray and Akkaya as we mentioned above. The waters, benthic sediments, and living species in the basin were contaminated by toxic chemicals such as Pb, Ni, Al, As, nonylphenol, trichloromethane, and benzo [a] pyrene derived from industry, agriculture, and sewage. These chemicals were a major threat not only to aquatic gastropods, other aquatic species, and terrestrial species in the surroundings, but also to humans through bioaccumulation and trophic transfer. In addition, excessive water withdrawal, fluctuating water depth, and invasive or native fish species introduced for fishing followed by drought periods hampered the gastropod survival in KCB.

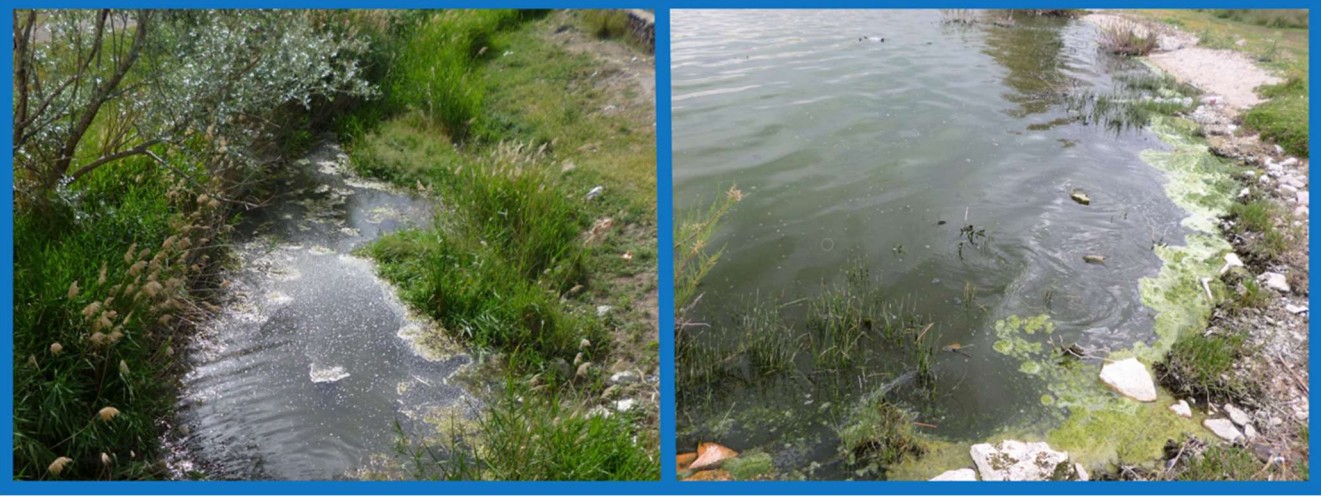

**Figure 7.** Algal blooms in Aksaray Input (on the **left**), and Akkaya Dam (on the **right**); photographed by Ahmet Altındağ on May 2014.

Habitat diversity induced by sediment heterogeneity, composition, and complexity of the macrophyte community increases the species richness [63–65]. According to the intermediate disturbance hypothesis, moderate levels of disturbance create conditions that foster greater species diversity than low or high levels of disturbance [66]. As can be seen in Figure 8, the macrophyte flora and the mesotrophic structure of Beyşehir supported the richness, and diversity of gastropods. Our previous survey in 2021 showed that there was an excessive water withdrawal in Beyşehir Lake. Hundreds of dead animals, including molluscs, covered the lakeshore following water withdrawal (Figure 9). Obviously, the ecosystem services of the lake were neglected. Thus, a revised management plan including regulations concerning irrigation water supply, fisheries, agriculture, reed cutting, habitat fragmentation, tourist boat trips, and beaches is urgently needed.

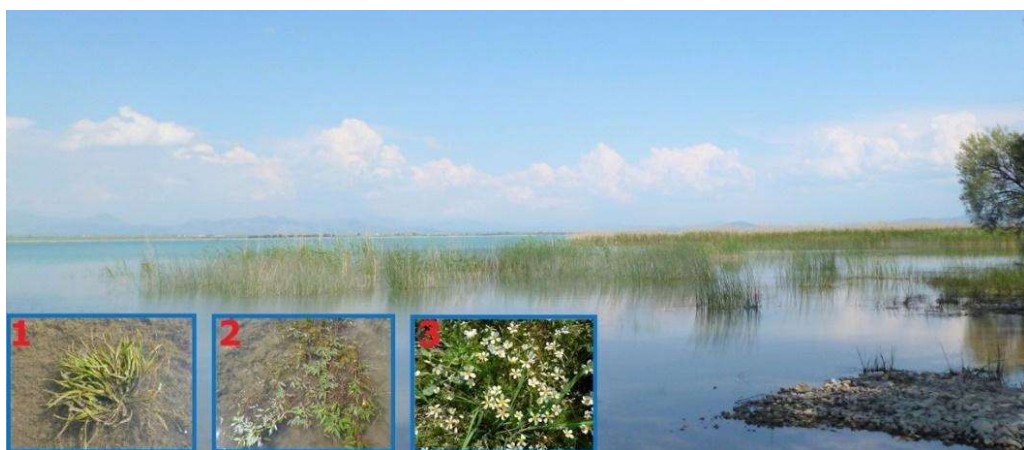

**Figure 8.** Beyşehir Lake shore with phragmites, and macrophytes (1: *Potamogeton natans*, 2: *Sagittaria subulata*, 3: *Ranunculus aquatilis*); photographed by Ahmet Altındağ on May 2014; produced by Burçin Aşkım Gümüş on April 2022.

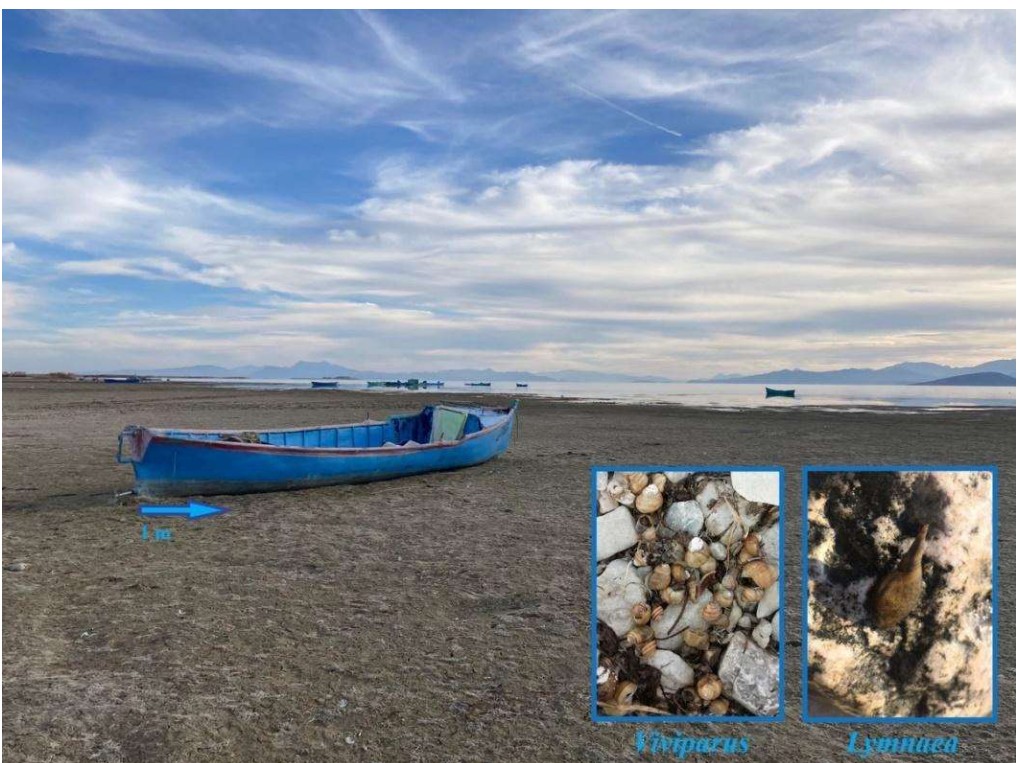

**Figure 9.** Flock mortality in freshwater snails following excessive water withdrawal in Beyşehir Lake (*Viviparus viviparus*, 4.5 cm width; 1 survivor of *Lymnaea stagnalis*, 5 cm height; small fishing boat, 6 m length; photographed by Burçin Aşkım Gümüş on November 2021).

Whether it is a prosobranch or pulmonate, there is a great variety of animals feeding on snails. They have excellent conchological and behavioural antipredator adaptations by taking advantage of spatial–temporal refuges and microhabitats [67]. What happens if there are too many non-native, omnivorous, and detritivorous fish in the territory? *Carassius gibelio*, *Pseudorasbora parva*, *Sander lucioperca*, *Atherina boyeri*, *Gambusia affinis*, and *Cyprinus carpio* have already invaded the humanly disturbed waters of KCB [12]. Apparently, they were one of the greatest threats to gastropods in KCB as well as chemical contamination.

In the basin, we identified two endemic species. In Çeltik, there was just one living specimen of *Theodoxus anatolicus,* whereas in Beyşehir, there were seven specimens of *Bithynia pseudemmericia.* This indicates that their IUCN [68,69] categories require a thorough

revision following the publication of this study, as they might well reach the level of being critically endangered (CR).

## 5. Conclusions

The results of water quality assessments in 13 freshwater bodies of KCB are important for Türkiye. We observed and determined the current situation of malacofauna in the study area. Despite the lack of comparative studies previously published, the first author personally observed the decline in the last decade, but that there was no quantitative data available. We have experienced the dramatic consequences of disturbing freshwater with various environmental stressors for decades at high levels in Türkiye and worldwide. As mentioned by Lydeard et al. [70], the loss and decline of charismatic vertebrates, such as polar bears, and invertebrates, such as those of butterflies and corals, attract scientific and public interest. However, tiny, fragile gastropods with a maximum size of 5.5 cm that live in muddy or silty bottoms of freshwaters are not of interest. Freshwater gastropods comprise 5% of the world's gastropod fauna and unfortunately, they account for 20% of the recorded mollusc extinctions. In Europe, the situation is much more dramatic. Although the freshwater gastropods represent about 94% of the total number of freshwater mollusc species, 43.7% of the species are considered as threatened, with at least 12.8% of them being critically endangered, 10.5% endangered and 20.4% vulnerable [71,72]. The conservation of freshwater gastropods is linked with the conservation of biodiversity and the ecosystem structure of water. It has to be stressed that ensuring the accessibility of drinkable and healthy freshwater for the (human) public is a major task to be undertaken. Sensitive animals, such as freshwater molluscs, are indicator organisms, and the strong declining populations as revealed by our investigation clearly show that immediate action has to be taken. As a result, the gastropods are the top conservation priority for achieving a sustainable world through healthy aquatic environments [73].

*Suggestions*

In light of our results, we suggest strong restrictions in KCB: I—stop the influx of nutrients and heavy metal loads from agriculture, industry, and sewage; II—stop the introduction of non-native fishes; and III—seriously reduce water withdrawal and habitat deterioration by anthropogenic activities. Furthermore, the permanent biomonitoring of plant and animal populations, including molluscs, should be implemented in the basin to monitor the future development of its biodiversity.

**Author Contributions:** All authors contributed to the study conception, methodology, and design. B.A.G. performed identifications, measurements, and ecological evaluation of the gastropods, prepared the original draft of the manuscript; wrote, reviewed, and edited the manuscript. All authors commented on previous versions of the manuscript. P.G. was responsible for the statistical analyses, and interpretation of data, and A.A. was the project leader, and was responsible for field studies. All authors have read and agreed to the published version of the manuscript.

**Funding:** This study was part of the project funded by the Republic of Türkiye, Ministry of Forestry and Water Affairs (SGYM/ 33.01.32.00/ 2014).

**Institutional Review Board Statement:** This article does not contain any studies with human participants, vertebrates, crustaceans, and cephalopods performed by any of the authors.

**Acknowledgments:** This study is dedicated to the late Nuray Kayıkçı who was a nature lover, and beloved mother of Burçin Aşkım Gümüş. Our grateful thanks go to the authorities of the Republic of Türkiye, Ministry of Forestry and Water Affairs, the members of our project team, and our special thanks to Nesil Ertorun and Nurdan Terzi for their support in collecting molluscs, and to Eike Neubert for his valuable suggestions in improving our manuscript and English editing.

**Conflicts of Interest:** The authors declare that there is no conflict of interest.

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
