# Peer review of "Towards a Sustainable World: Diversity of Freshwater Gastropods in Relation to Environmental Factors—A Case in the Konya Closed Basin, Türkiye"

_diversity, doi:10.3390/d14110934_

Round 1

Reviewer 1 Report

Review of:

Title: Towards a sustainable world: water quality assessments based on freshwater gastropods — a case in the Konya closed basin, Türkiye

Authors: Burçin AÅŸkım GümüÅŸ, Pınar Gürbüzer, and Ahmet AltındaÄŸ

The authors report on the water quality in 13 freshwater bodies of Konya closed Basin in Turkey and determined the current situation of the freshwater gastropods in this area.

The paper is well done but the used species names are problematic. The authors use for instance Peregriana peregra or Ampullaceana balthica, used by Russian authors only. The authors wrote that they used “Schütt [35], Welter-Schultes [36], and Glöer [37]” for species identification” but these authors do not deal with these names. In addition the book of Glöer [37] concerns Germany only. Instead of this book the authors can use: Glöer, P. 2019-2022: The Freshwater Gastropods of the West- Palaearctis, vols 1-3. It would be better the authors use the species names used in Gürlek et al. [reference 1], the most modern check-list for Turkey.

After changing the names the ms should be published.

Author Response

Dear Colleague,

Thank you for your suggestions, and contribution.

Kind regards.

Reviewer 2 Report

Manuscript ID: diversity-1948134

Title: Towards a sustainable world: water quality assessments based on freshwater gastropods — a case in the Konya closed basin, Türkiye

Authors: Burçin AÅŸkım GümüÅŸ, Pınar Gürbüzer, Ahmet AltındaÄŸ

Submitted to section: Biodiversity Loss & Dynamics,

This is a good ecological study that constitutes a revision of the topic literature, adds some new data to the knowledge of mollusks, and required a lot of work from the authors in the field and in the laboratory. The authors revealed in their study that most of the gastropods in KCB are relatively tolerant to biodegradable pollution. However, there is a strong observed decline in population size requiring intensive future monitoring. The research results are limited to the area of Konya closed basin in Türkiye; however, they may constitute important data and serve as a model in this type of research. Before publication, this manuscript needs deep revision. Below some comments and corrections are placed, which I hope will help authors to improve their paper.

1.      I suggest changing the title of the manuscript, e.g., Towards a sustainable world: diversity of  freshwater gastropods in relation to environmental factors — a case in the Konya closed basin, Türkiye

2.      Line 155: The stations in the study area…? Or rather the sampling sites, sampling points. I would suggest changing the “stations, sampling stations” to “ sampling sites” or “ sampling points,”- which is usually used in the ecological studies (the first) and monitoring studies ( the first and the second term).

3.      “…in lotic stations, where the water current was fast, and the bottom was stony, a metal framed, and rectangular net with 0.5 mm mesh size, and a Ponar grab were used; 2) in lentic stations, where the water was deep, stagnant, and muddy, an Ekman-Birge grab with girdle sets with 10 mm, 5 mm, 2.5 mm, and 0.5 mm diameters were used; 3) in shallow lotic and lentic stations from the supra littoral and littoral zones, hand zooplankton nets, girdles, pincers, and hand collecting were used (Figure 2)…”

- the methods of sampling used by the authors were different. Quantitative methods of sampling enable to compare the density and the diversity between sampling sites, but what about the “ hand zooplankton nets, girdles, pincers, and hand collecting were used”- these methods are not quantitative, and how authors compared density and diversity between sampling sites located in shallow lentic waters with those located in lentic and deep waters and with the lotic once when the methods of sampling used gave incomparable results?.

4.      Was sampling “by hand” done from a specific surface of the substrate, the bottom? If yes, then they would be comparable. This part of the manuscript should be supplemented with lacking information.

5.      2.5. Statistical Methods;  Why was the pH not log transformed?

6.      Lines 228-229 „…According to detrended correspondence analysis (DCA), the data showed unimodal behaviour…”- behavior must be changed into “distribution.”

-The length of the gradient in the analysis should be given in the text of the statistical methods

7.      Table 3:. “Species list, number of the identified specimen (individual m-2)...”

- the density must have been calculated from the samples done with different equipment- this must be given in methods, how authors calculated density (ind./m2) in samples collected by hand or by zooplankton net? There is a lack of information in Materials and Methods. Something is not clear here. How the sampling was done needs detailed explanation, e.g., using Eckman grab, we usually do not take samples from the area of 1m2, but calculate the density taking into account the Eckman grab area, then, how the density was assessed using this (and other) methods?

8.      3.2.2. CCA results of lotic stations

- Why do authors reference the results with the literature in this part of the text? In the “Results,” only the results obtained by authors should be presented; then, in the “Discussion,” authors need to relate their results with the literature data obtained by other scientists in a wide context.

9.      Figure 5. All abbreviations need to be explained In the figure caption.

10.  Cluster analysis and MDS: In my opinion, one cluster including every sampling site (lotic, lentic, and others) should be made, then we would see if the composition of gastropods were similar in lentic waters, etc. this would give information between which environments are the faunistic similarities. The same should be done in the case of the MDS; one figure with all of the sampling sites should be presented and, if necessary, marked with circles places located close-to-each depending on the relations and the results from such an analysis. This means that section Results should be rewritten. Authors in their manuscript revision should also remember to use the past tense and not the present tense, e.g., “snails were” instead of “snails are”.

11.  Lines 436-437 „Besides the first author would like to draw attention about the excessive water withdraw in BeyÅŸehir Lake during her expedition on 08 November 2021.”

-This sentence should be removed from the manuscript or rewritten, e.g., “ Our previous study on November 2021 showed that………..”.

12.  All photographs need to have an author name in their caption, “photographed by the first author on 08 November 2021” needs to be changed to author name and surname + if needed “on 08 November 2021”.

Author Response

Dear Colleague,

Thank you for your suggestions and contributions.

Kind regards.

Round 2

Reviewer 1 Report

The ms can be published

Reviewer 2 Report

This manuscript is very interesting and fits the requirements of scientific journals like DIVERSITY. The presented version of this paper, corrected according to all suggestions, is now well prepared, and in my opinion, it can be published. I have no more comments on this manuscript.